# Prevalence of clinically significant refractive error in children in Europe: Systematic review and meta-analysis

Megan Doyle[1‡], Aoife O'Donnell[2‡], Síofra Harrington[1], Veronica O'Dwyer[1], Michael Moore[1]*

1 Centre for Eye Research Ireland, School of Physics, Clinical & Optometric Sciences, Technological University Dublin, Dublin, Ireland, 2 School of Medicine, Ulster University, Derry~Londonderry, Northern Ireland

‡ MD and AOD are joint first authors to this work.
* michael.moore@tudublin.ie

## Abstract

### Purpose

This meta-analysis estimated clinically significant refractive error prevalence in European children and examined variations based on cycloplegic versus non-cycloplegic refraction, age and sex.

### Methods

A systematic review (PubMed, Cochrane Library, EMBASE, PROSPERO ID: CRD42022322608) identified studies (January 2000 – June 2024) reporting myopia (≤−0.50D), hyperopia (≥+2.00D) and astigmatism (≥1.00 DC) prevalence in children aged 4–17 years in Europe. Twenty-six studies from 14 countries (n = 37,282) were included. Pooled prevalence estimates (EPP) were calculated using a random-effects model with Freeman-Tukey double-arcsine transformation.

### Results

Estimated pooled prevalences were refractive error: 17.90% (95% CI: 13.67–22.56), myopia: 14.31% (95% CI: 8.89–20.74; 24 studies), hyperopia: 10.20% (95% CI: 6.51–14.59; 11 studies), and astigmatism: 10.26% (95% CI: 5.83–15.74, eight studies). Cycloplegic and non-cycloplegic prevalence did not differ significantly (myopia: $\chi^2 = 0.08$, p = 0.78; hyperopia: $\chi^2 = 0.29$, p = 0.59). Myopia prevalence increased with age (6.17% at 4–9 years, 16.66% at 14 + years), while hyperopia declined (14.27% at 4–9 years, 7.04% at 10–13 years). Astigmatism remained stable; however, studies did not report its co-occurrence with myopia or hyperopia, limiting insights into its combined burden. No significant sex differences were observed. Data from 27 European countries were unavailable, limiting regional comparisons.

**Data availability statement:** All relevant data are within the paper and its Supporting Information files.

**Funding:** The author(s) received no specific funding for this work.

**Competing interests:** I have read the journal's policy and the authors of this manuscript have the following competing interests: MM is a consultant for Thea Pharmaceuticals.

## Conclusions

Approximately 18% of European children have clinically significant refractive error, with myopia increasing with age and hyperopia decreasing. Myopia prevalence was highest in Russia, hyperopia in Denmark, and astigmatism in Northern Ireland. Further studies reporting cycloplegic prevalences with improved geographical representation and more granular reporting – particularly of astigmatism and its co-occurrence with other refractive errors are needed for better comparability and management of refractive errors in European children.

## Introduction

Uncorrected refractive error is a leading cause of visual impairment worldwide and a growing public health concern, particularly in children. Uncorrected refractive error not only limits educational and developmental potential but also increases the risk of amblyopia and, at higher magnitudes, irreversible sight-threatening complications such as myopic macular degeneration [1,2]. The global economic burden of visual impairment is substantial and far-reaching—estimated at $408.5 billion annually— with uncorrected refractive error contributing significantly [3].

While severe refractive error carries greater individual risk, the broader public health burden stems from the vast number of people with low to moderate levels of refractive error, particularly myopia. Despite predictions that nearly half the world's population will be myopic by 2050 [4], robust data on paediatric refractive error in Europe remain surprisingly limited. Previous global reviews have included only a handful of European studies, often focusing exclusively on myopia or neglecting hyperopia and astigmatism altogether [5,6].

Understanding prevalence is not just a statistical exercise—it is essential for designing screening programmes, allocating resources, and preventing future visual impairment. Yet, inconsistent use of cycloplegia further complicates the comparability of findings. Cycloplegic refraction is the gold standard for epidemiological studies in children, critical for accurately identifying latent hyperopia and avoiding overestimating myopia [7,8]. However, its use is restricted in several European countries [9], raising concerns about methodological consistency across studies.

This review and meta-analysis aim to address these gaps by (1) providing updated estimates of the prevalence of clinically significant refractive error (myopia, hyperopia, and astigmatism) in European children aged 4–17 years, (2) comparing prevalence rates between cycloplegic and non-cycloplegic studies, and (3) analysing variations by age and sex.

## Materials and methods

The current systematic review procedure adhered to the Preferred Reporting Items for Systematic Reviews and Meta-Analyses (PRISMA-P) guidelines and reported as per the meta-analyses of observational studies in epidemiology (MOOSE) checklist (S1 and S2 Tables, respectively). The review was registered on PROSPERO

(University of York, https://www.crd.york.ac.uk/prospero/) (ID: CRD42022322608). Meta-analysis protocol is described in S1 Appendix.

## Literature search strategy

Studies from European countries were extracted from PubMed, Cochrane Library and EMBASE databases. To maximise the available evidence, studies published from 1 January 2000–30 September 2024 were included. This broad timeframe was consistent with previous prevalence meta-analyses and ensured the inclusion of recent, methodologically robust studies while allowing adequate data coverage for European children, where studies remain limited [5,6,10]. Myopia prevalence and refractive error patterns have changed over the past two decades due to lifestyle shifts, increased near work, and reduced outdoor time [6]. Including studies from 2000 onward allows for the analysis of modern trends in paediatric refractive error prevalence. Two independent researchers (MD, AOD) performed the literature search following the same protocol between March 2022 and September 2024. The PubMed, Cochrane Library and EMBASE databases were searched using the search terms and Boolean operators for each country, as shown in Table 1.

Studies were initially assessed for relevance based on their title and abstract. Accepted papers were collated into a list country-by-country for review. The World Health Organisation (WHO) region classification for European countries was used (https://www.who.int/countries) [11].

## Study screening and appraisal

Articles reporting duplicate data were removed. Each study was scored out of 9 per the Joanna Briggs Institute (JBI) criteria (S3 Table). Any disagreements were settled by a third party (SH). The Grades of Recommendations Assessment, Development and Evaluation (GRADE) system was used to rate the quality of evidence of included studies. Observational studies were classified as low-quality evidence and could be further downgraded based on five criteria, following the procedure detailed in Balshem et al. [12]. Studies were then rated as having high, moderate, low, or very low-quality evidence. Excluded studies and the reasons for exclusion are in S5 Table. Data from each study were added to a prefabricated Microsoft Excel template, including authors, publication year, study location, urban or rural setting, sample size, sex distribution, age groups, refractive error definitions, cycloplegia use, and prevalence values. Fig 1 presents a flowchart of the paper selection process for systematic review following the literature search.

## Inclusion and exclusion criteria

The inclusion criteria for the systematic review were:

(1)  Original research papers published between the years 2000–2024 (inclusive),

(2)  Studies were conducted in European countries,

(3)  Refractive error prevalence was reported in categories (myopia, hyperopia and astigmatism),

**Table 1.  Search terms used for the Pubmed, Cochrane Library and EMBASE databases for each country in Europe.**

| [Country] | AND | "Refractive error" |
|-----------|-----|--------------------|
| [Country] | AND | "Myopia" |
| [Country] | AND | "Hyperopia" |
| [Country] | AND | "Astigmatism" |
| [Country] | AND | "Ametropia" |

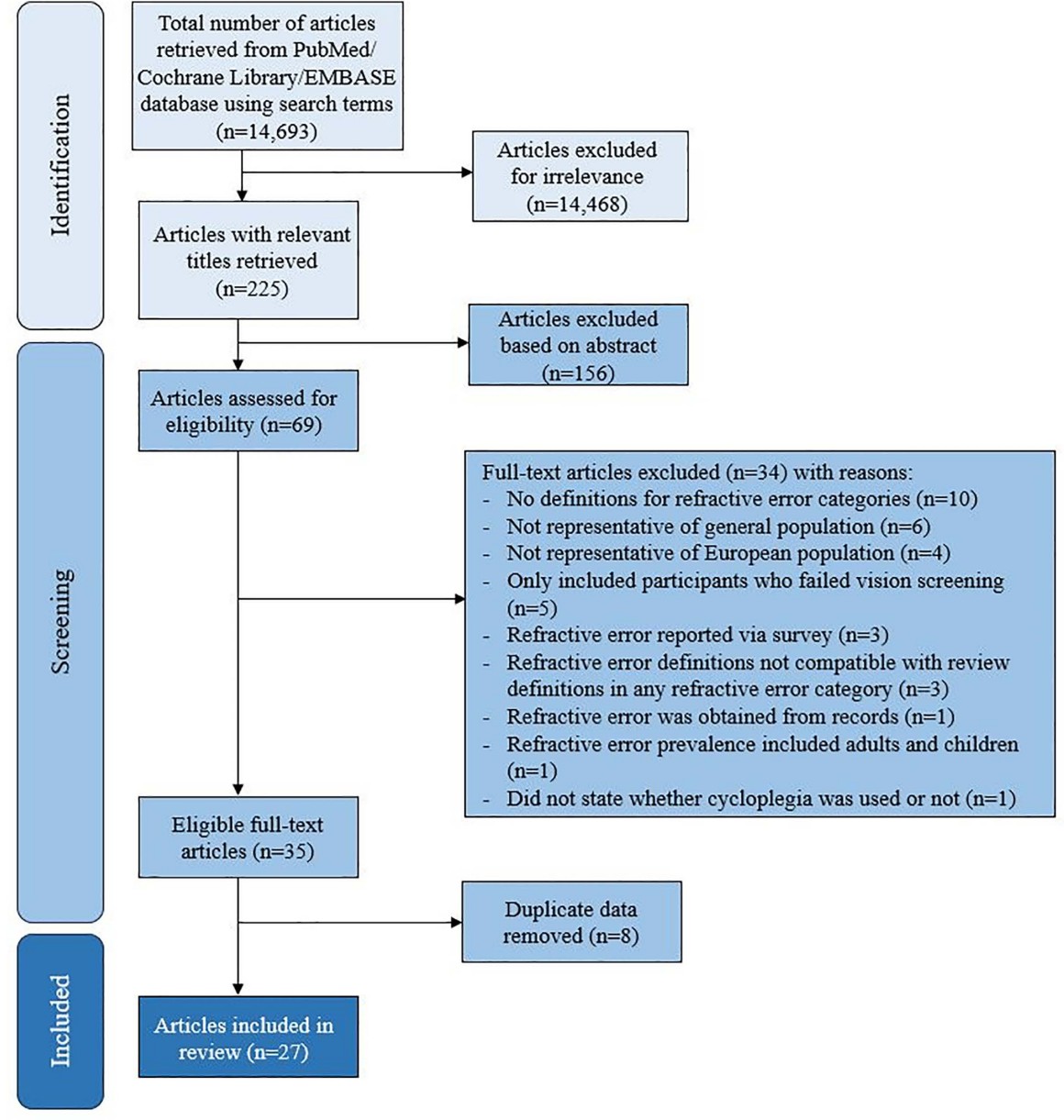

**Fig 1. PRISMA flowchart of steps in identifying suitable studies.**

(4)  Definitions were provided for each refractive error category,

(5)  Participants in each study were between the ages of 4–17 inclusive,

(6)  Sample size must be stated,

(7)  Studies must have specified whether cycloplegic agents were employed,

(8)  Studies must be representative of the general paediatric population; thus, excluding those focusing on ocular or systemic disease states, or clinic-based studies,

(9) Refractive error must not be self-reported; studies that used participant surveys to determine the prevalence of refractive error were excluded.

The literature search included studies from the following countries: Albania, Andorra, Armenia, Austria, Azerbaijan, Belarus, Belgium, Bosnia & Herzegovina, Bulgaria, Croatia, Cyprus, Czechia, Denmark, Estonia, Finland, France, Georgia, Germany, Greece, Hungary, Iceland, Ireland, Israel, Italy, Latvia, Lithuania, Luxembourg, Malta, Monaco, Montenegro, Netherlands, North Macedonia, Norway, Poland, Portugal, Republic of Moldova, Romania, Russian Federation, San Marino, Serbia, Slovakia, Slovenia, Spain, Sweden, Switzerland, Ukraine, United Kingdom (and its components: England, Scotland, Wales and Northern Ireland (NI)).

Unpublished research and abstract-only data were not included due to the inability to assess study quality. Accessible studies written in a language other than English were translated online and assessed for eligibility.

## Definitions

Refractive error prevalence pooled studies reporting myopia and hyperopia. Astigmatism was excluded from the refractive error prevalence analysis due to inadequate reporting and methodological limitations. Some studies reported separate prevalence values for myopia, hyperopia and astigmatism, but did not specify how many participants had both astigmatism and myopia or hyperopia. This lack of distinction could lead to double counting when calculating total clinically significant refractive error prevalence, as astigmatism often co-exists with myopia or hyperopia [13]. Instead, astigmatism prevalence was analysed separately. Clinically significant myopia was defined as spherical equivalent refraction (SER) ≤−0.50D, hyperopia as SER ≥+2.00D, and astigmatism as ≥1.00 DC. Myopia and hyperopia definitions followed the Refractive Error Study in Children (RESC) guidelines [14].

Within eligible studies, variations in hyperopia and astigmatism definitions resulted in the exclusion of some studies from specific refractive error analyses. For example, a study's myopia prevalence data were included, but its hyperopia data were excluded if it did not use the ≥+2.00D required for inclusion. Subgroup analyses examined the effect of differing definitions on prevalence estimates. For hyperopia, studies which used the RESC definition (≥+2.00D) were compared to studies which used lower thresholds (+0.50 ≤ SER<+2.00D). Studies which defined astigmatism as ≥1.00 DC were compared to those which used a lower threshold of between 0.50 DC and <1.00DC.

## Data analysis

Statistical analyses were performed with R version 4.1.2 (The R Project for Statistical Computing, r-project.org, 2021), an open-source statistical analysis software using the *meta* package. Individual study proportions and estimated pooled prevalences (EPP) were assessed and reported with 95% confidence intervals for each refractive error category. The Freeman-Tukey double arcsine transformation was applied to minimise the effect of studies with extremely high or low prevalence estimates on the overall pooled estimates. Cochran Q statistics and $I^2$ were used to assess heterogeneity between studies. The random effects model was chosen to analyse pooled estimates to account for heterogeneity between studies. Leave-one-out analyses, Baujat plots and studentised residuals were used to identify potential outliers and quantify each study's effect on the pooled proportion. Publication bias was evaluated using funnel plots and Egger's test.

Subgroup analyses were performed to determine prevalence estimates by cycloplegic status, age, sex and refractive error definition. Prevalence values were compared between cycloplegic and non-cycloplegic studies (Table 2). Age was separated into three groups: 4–9 years, 10–13 years, and 14+years.

## Results

Twenty-seven studies from 14 European countries were included in the principal analyses of myopia, hyperopia and astigmatism prevalence, with 37,282 participants. Table 2 presents the summary characteristics of the included studies.

**Table 2. Summary characteristics of included studies (n = 37,282 participants). The participant count does not include O'Donoghue (2010) [15], O'Donoghue (2015) [20], or McCullough (2016) [17], as these papers all report on the Northern Ireland Childhood Errors of Refraction study, and the participant count for this dataset is taken from O'Donoghue (2011) [16].**

| First Author (Year) | Country | Urban/ Rural (U/R) | Cyclo (Y/N) | Myopia Definition (D) | Hyperopia Definition (D) | Astigmatism Definition (D) | Age (years) | Sample Size | Male n (%) | Female n (%) | Myopia % | Hyperopia % | Astigmatism % |
|---|---|---|---|---|---|---|---|---|---|---|---|---|---|
| Villarreal (2000) [21] | Sweden | U | Y | ≤−0.50 | ≥+1.00 | ≥1.50 | 12-13 | 1045 | 532 (50.9) | 513 (49.1) | 49.67 | 8.4 | 5.2 |
| Gronlund (2006) [22] | Sweden | U | Y | ≤−0.50 | ≥+2.00 | ≥1.00 and ≥0.75 | 4-15 | 143 | 76 (53.1) | 67 (46.9) | 6.29 | 9.09 | 8.39 and 22.38 |
| Czepita (2007) [23] | Poland | U | Y | ≤−0.50 | ≥+1.00 | NR | 6-17 | 4235 | NR | NR | 12.44 | 13.48 | NR |
| Hendricks (2007) [24] | Netherlands | NR | N | <−0.50 | >+0.50 | >0.25 | 11-13 | 487 | 235 (48.3) | 252 (51.7) | 15.00 | 12.00 | 33.00 |
| Abdi (2008) [25] | Sweden | U/R | Y | ≤−0.50 | ≥+1.25 | >1.25 | 6-7, 10-12, 14-16 | 216 | 105 (48.6) | 111 (51.4) | 9.26 | 9.72 | 9.26 |
| Williams (2008) [26] | England | R | N | NR | ≥+2.00 | NR | 7 | 7538 | 3828 (50.8) | 3710 (49.2) | NR | 4.8 | NR |
| Plainis (2009) [19] | Bulgaria | U | N | ≤−0.75 | NR | >0.75 | 10-15 | 310 | 151 (48.7) | 159 (51.3) | 13.5 | NR | 9.7 |
| Plainis (2009) [19] | Greece | U | N | ≤−0.75 | NR | >0.75 | 10-15 | 588 | 313 (53.2) | 275 (46.8) | 37.2 | NR | 16.8 |
| O'Donoghue (2010) [15] | Northern Ireland | NR | Y | ≤−0.50 | ≥+2.00 | NR | 6-7, 12-13 | 1051 | 527 (50.1) | 524 (49.9) | 12.18 | 19.03 | NR |
| Rudnicka (2010) [27] | England | U | N | ≤−0.50 | NR | NR | 10-11 | 1029 | 496 (48.2) | 533 (51.8) | 11.9 | NR | NR |
| Logan (2011) [28] | England | U | Y | ≤−0.50 | ≥+2.00 | NR | 6-7, 12-13 | 596 | NR | NR | 18.46 | 9.23 | NR |
| O'Donoghue (2011) [16] | Northern Ireland | NR | Y | NR | NR | ≥1.00 | 6-7, 12-13 | 1053 | 528 (50.1) | 525 (49.9) | NR | NR | 21.46 |
| Polling (2012) [29] | Poland | R | Y | ≤−0.50 | ≥+2.00 | NR | 4-12 | 422 | NR | NR | 4.27 | 10.19 | NR |
| Larsson (2015) [30] | Sweden | NR | Y | ≤−0.50 | ≥+2.00 | ≥1.00 | 10 | 217 | 104 (47.9) | 113 (52.1) | 7.83 | 3.69 | 4.15 |
| Lundberg (2015) [31] | Denmark | U | Y | ≤−0.50 | NR | NR | 14-17 | 307 | 161 (52.4) | 146 (47.6) | 17.92 | NR | NR |
| O'Donoghue (2015) [20] | Northern Ireland | U/R | Y | NR | NR | ≥1.00 | 9-10, 15-16 | 724 | NR | NR | NR | NR | 17.27 |
| McCullough (2016) [17] | Northern Ireland | U/R | Y | ≤−0.50 | ≥+2.00 | NR | 12-13 | 212 | 105 (49.5) | 107 (50.5) | 14.6 | 14.2 | NR |
| Tideman (2017) [32] | Netherlands | U | Y | ≤−0.50 | NR | NR | 6 | 5711 | 2862 (50.1) | 2849 (49.9) | 2.40 | NR | NR |
| Popović-Beganović (2018) [33] | Bosnia & Herzegovina | U | Y | ≤−0.50 | ≥+2.00 | ≥0.75 | 7-16 | 997 | NR | NR | 8.32 | 4.01 | 13.34 |
| Hagen (2018) [34] | Norway | U | Y | ≤−0.50 | ≥+0.50 | NR | 16 | 246 | 107 (43.5) | 139 (56.5) | 10.98 | 57.72 | NR |
| Sandfeld (2018) [35] | Denmark | U | Y | <−0.50 | >+2.00 | >1.00 | 4-7 | 445 | 231 (51.9) | 214 (48.1) | 0.00 | 27.6 | 4.27 |
| Harrington (2019) [36] | Ireland | U/R | Y | ≤−0.50 | ≥+2.00 | ≥1.00 | 6-7, 12-13 | 1626 | 881 (54.2) | 745 (45.8) | NR | 16.11 | 17.40 |
| Slaveykov (2020) [37] | Bulgaria | U | N | ≤−0.50 | ≥+2.00 | ≥1.00 | 4-6 | 434 | 208 (47.9) | 226 (52.1) | 7.83 | 11.98 | 8.4 |
| Demir (2021) [38] | Sweden | R | Y | ≤−0.50 | ≥+0.75 | NR | 8-16 | 128 | 58 (45.3) | 70 (54.7) | 10.16 | 47.66 | NR |

*(Continued)*

**Table 2.** (Continued)

| First Author (Year) | Country | Urban/ Rural (U/R) | Cyclo (Y/N) | Myopia Defini- tion (D) | Hyperopia Definition (D) | Astigma- tism Defini- tion (D) | Age (years) | Sam- ple Size | Male n (%) | Female n (%) | Myo- pia % | Hyper- opia % | Astig- matism % |
|---|---|---|---|---|---|---|---|---|---|---|---|---|---|
| Alvarez-Peregrina (2021) [39] | Spain | NR | N | ≤−0.50 | >+0.50 | NR | 5-7 | 1601 | 792 (49.5) | 809 (50.5) | 19.43 | NR | NR |
| Dragomirova (2022) [40] | Bulgaria | U/R | N | ≤−0.75 | NR | NR | 6-10, 11-15 | 1401 | 668 (47.7) | 733 (52.3) | 16.85 | NR | NR |
| Martinez-Perez (2022) [41] | Portugal | U | N | ≤−0.50 | ≥+0.50 | NR | 6-12 | 252 | 127 (50.4) | 125 (49.6) | 9.5 | 56.3 | NR |
| Monika (2023) [42] | Poland | U | Y | ≤−0.50 | ≥+2.00 | ≥0.75 | 8 | 1518 | 791 (52.1) | 727 (47.9) | 16.60 | 5.80 | 10.61 |
| Bikbov (2023) [43] | Russia | U | Y | ≤−0.50 | >+0.50 | NR | 6-17 | 4737 | 2328 (49.1) | 2409 (50.9) | 46.17 | 11.1 | NR |

*Not reported (NR), Dioptre (D), number of participants (n), Yes (Y), No (N), Urban (U), Rural (R).*

Five studies were conducted in Sweden, three each in Poland, the United Kingdom (England specifically), and Bulgaria, two each in the Netherlands, Northern Ireland (NI) and Denmark, and one each in Ireland, Greece, Bosnia & Herzegovina, Norway, Spain, Portugal, and Russia. Eighteen studies used cycloplegia, and nine did not. Twenty studies were school-based, and seven studies were population-based. Some studies were included more than once between the three refractive error analyses. Four papers from the Northern Ireland Childhood Errors of Refraction (NICER) study were included. Two reporting baseline data were in the principal analysis, while two reporting longitudinal data were only included in the age subgroup analyses. Two baseline prevalence papers from the NICER dataset were published: myopia and hyperopia [15], and astigmatism [16]. Their sample sizes were not double-counted in the current review. Only baseline NICER data were included in the principal prevalence analysis. Two additional studies reporting longitudinal myopia and hyperopia prevalence [17], and astigmatism prevalence [18], were included in the age subgroup analysis but excluded from the overall analysis due to their longitudinal nature. Additionally, one paper included participants from both Bulgaria and Greece [19]; the current review treats these as separate studies as they are independent datasets from different countries.

All included studies achieved a score of five or higher using the JBI Critical Appraisal tool, with the highest score being nine. Additional details on the JBI appraisal of included studies are presented in S3 Table. The quality of evidence was initially rated as "low" according to the GRADE Working Group protocol (S4 Table) due to the observational nature of the included studies [12]. The overall GRADE assessment classified the evidence as "very low" quality due to concerns about bias, including limited geographical coverage, underreporting of older age groups, less frequent reporting of hyperopia and astigmatism prevalence, and inconsistent refractive error definitions across studies.

Leave one out analysis did not identify any influential outliers (S1 Fig). There was insignificant asymmetry in the funnel plot, as confirmed by Egger's test (z = −1.0148, p = 0.3102), indicating no publication bias (Fig 2).

### Refractive error prevalence

The EPP of significant refractive error (including myopia and hyperopia prevalence studies only) in children in Europe was 17.90% (95% CI: 13.67–22.56, $I^2$ = 99.2%, Q = 5951.44 [df = 25], Fig 3). If studies reporting both myopia and hyperopia alongside astigmatism were included, the prevalence was 20.81% (95% CI: 15.27–26.97, I2 = 99.5%, Q = 7539.73 [df = 25]).

Myopia, hyperopia and astigmatism prevalences were 14.31% (95% CI: 8.89–20.74, $I^2$ = 99.5%, Q = 4663.93 [df = 23]; 24 studies), 10.20% (95% CI: 6.51–14.59, $I^2$ = 98.1%, Q = 528.22 [df = 10]; 11 studies) and 10.26% (95% CI: 5.83–15.74, $I^2$ = 97.1%, Q = 239.45 [df = 7]; 8 studies) respectively (S2a-c Fig). Refractive error prevalence by country is depicted in

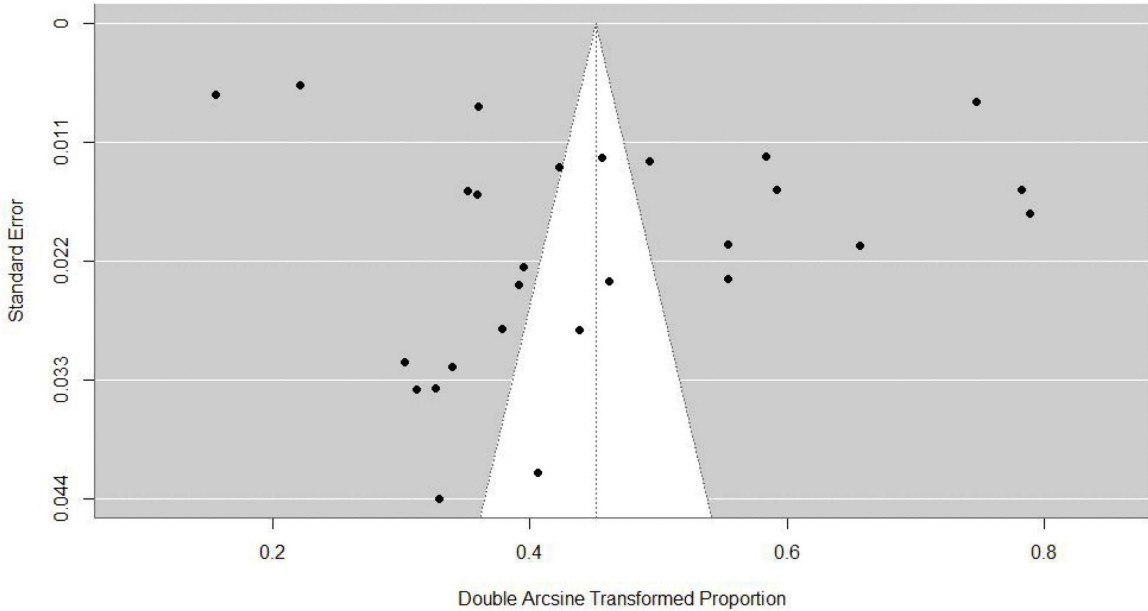

**Fig 2. Funnel plot of the included studies. Egger's test did not indicate significant publication bias (z = −1.0148, p = 0.3102).**

Fig 4. Myopia was most prevalent in Russia (46.17%, 95% CI: 44.73–47.58), followed by Greece (37.24%, 95% CI: 33.40–41.20) and Spain (19.43%, 95% CI: 17.51–21.39), and with the Netherlands having the lowest prevalence (7.30%, 95% CI: 0.34–21.83; S3(a) Fig). Hyperopia was most prevalent in Denmark (27.64%, 95% CI: 23.61–31.92), followed by Northern Ireland (18.99%, 95% CI: 16.68–21.42) and Ireland (16.11%, 95% CI: 14.37–17.94), with Bosnia and Herzegovina having the lowest prevalence (4.01%, 95% CI: 2.82–5.27; S3(b) Fig). Astigmatism was most prevalent in Northern Ireland (21.46%, 95% CI: 19.03–24.00), followed by Ireland (17.40%, 95% CI: 15.60–19.29) and Greece (17.18%, 95% CI: 14.23–20.34) with Denmark having the lowest prevalence (4.27%, 95% CI: 2.57–6.37; S3(c) Fig).

## Cycloplegic versus non-cycloplegic prevalence

The EPP of clinically significant refractive error in children in Europe was 19.17% (95% CI: 11.79–27.84; 17 studies) in cycloplegic studies and 15.50% (6.80–26.91; nine studies) in non-cycloplegic studies ($\chi^2 = 0.30$, p = 0.58). The EPP of myopia was 13.68% (95% CI: 7.18–21.83; 16 studies) in cycloplegic studies and 15.58% (95% CI: 6.32–27.95; seven studies) in non-cycloplegic studies ($\chi^2 = 0.08$, p = 0.78; Fig 5a). The EPP of hyperopia was 10.74% (95% CI: 6.52–15.85; nine studies) in cycloplegic studies and 7.96% (95% CI: 1.60–18.47; two studies) in non-cycloplegic studies ($\chi^2 = 0.29$, p = 0.59; Fig 5b). The EPP of astigmatism was 9.32% (95% CI: 4.28–15.99; 7 studies) in cycloplegic studies. Only one non-cycloplegic study reported astigmatism prevalence according to the definition for the current review, limiting EPP calculation (13.25%, 95% CI: 3.84–27.08).

## Age

Refractive error prevalence was lowest in the 4–9 year age group (14.31%, 95% CI: 8.75–20.93), increasing to its maximum in the 10–13 year group (19.14%, 95% CI: 12.55–26.71) and slightly decreasing in the 14+ year group (16.65%, 95% CI: 6.72–29.78). Myopia prevalence increased with age, more than doubling from 6.17% (95% CI: 2.61–11.03) in 4–9-year-olds to 16.42% (95% CI: 10.88–22.82) in 10–13-year-olds ($\chi^2 = 8.53$, p = 0.014), before stabilising in the 14+ age group (16.66%,

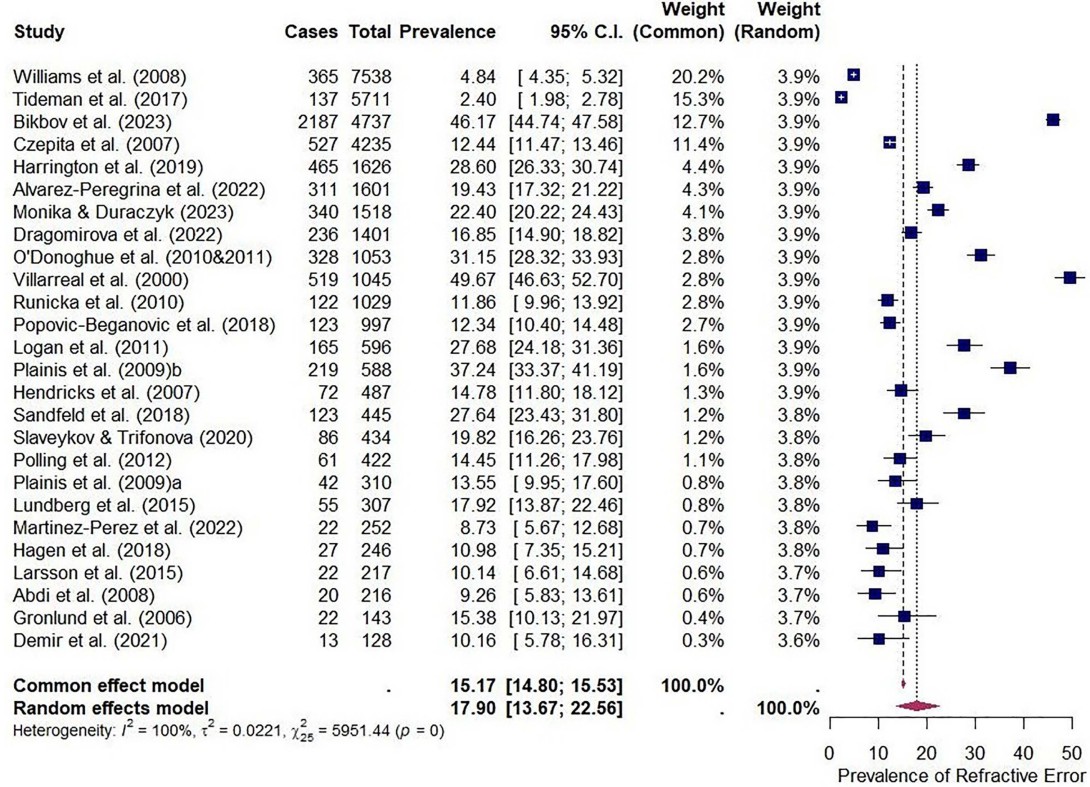

**Fig 3. Forest plot displaying the prevalence of clinically significant refractive error in 4-17-year-olds in Europe across 27 studies (37,282 participants, 14 countries, 18 cycloplegic, 9 non-cycloplegic).** Refractive error prevalence was 17.90% (95% CI: 13.67–22.56).

95% CI: 7.28–28.84). Hyperopia prevalence reduced with age from 14.27% (95% CI: 8.67–20.96) in 4–9-year-olds to 7.04% (95% CI: 2.34–13.82) in 10–13-year-olds ($\chi^2$=2.53, p=0.112). No studies reported hyperopia prevalence in the 14+year age group. Astigmatism prevalence remained stable between the 4–9 year (12.96%, 95% CI: 6.13–21.83) and 10–13 year (11.84%, 95% CI: 6.23–18.87) age groups, and was not significantly different at 14+years (17.48%, 95% CI: 14.03–21.23; $\chi^2$=0.37, p=0.83). Only one study reported astigmatism prevalence in the 14+year age group [18].

### Sex

There was no significant difference in refractive error prevalence between females (14.88%, 95% CI: 8.98–21.94) and males (14.02%, 95% CI: 8.28–20.95, $\chi^2$=0.03, p=0.85). Similarly, myopia prevalence (females: 15.67%, 95% CI: 8.71–24.18; males: 14.65%, 95% CI: 7.91–22.99, $\chi^2$=0.03, p=0.86) and hyperopia prevalence did not differ significantly by sex (females: 4.92%, 95% CI: 4.13–5.77; males: 5.17%, 95% CI: 4.38–6.03, $\chi^2$=0.18, p=0.67). Astigmatism prevalence by sex was only reported in one study, which defined astigmatism as >0.25DC. Including this study could artificially inflate prevalence, as such low levels of astigmatism are not clinically significant.

### Refractive error definition

Eight of the included studies provided hyperopia prevalence values using a definition lower than ≥+2.00D. A subgroup analysis showed that lower definitions of hyperopia (+0.50D ≤ SER<+2.00D) result in higher prevalence (28.47%, 95% CI: 20.46–37.22, $\chi^2$=14.47, p<0.01) than studies using a hyperopia definition of ≥+2.00D as reported above.

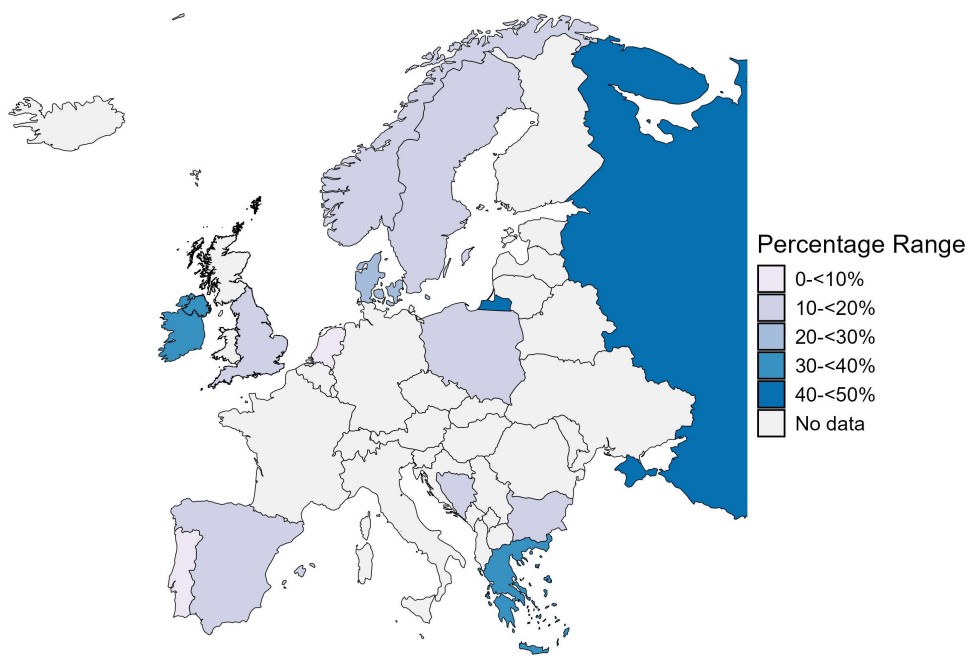

**Fig 4. Heat map of clinically significant paediatric refractive error prevalence ((myopia ≤ −0.50 and hyperopia ≤+2.00D) by country.** The EPP of clinically significant refractive error was 17.90% (95% CI: 13.67–22.56). Refractive error prevalence was highest in Russia (46.17%, 95% CI: 44.74–47.59), followed by Greece (37.24%, 95% CI: 33.37–41.19) and Northern Ireland (31.15%, 95% CI: 28.39–33.98)). Notably, only myopia prevalence was included from Russia and Greece, and there was a lack of clinically significant hyperopia or astigmatism data. Mapping data from http://www.naturale-arthdata.com/.

Astigmatism prevalence was higher in studies with lower definitions (between 0.50 DC and <1.00DC,16.29%, 95% CI: 8.87–25.41) compared to ≥1.00 DC (10.27%, 95% CI: 5.96–15.56). However, no statistically significant difference in prevalence values was found ($\chi^2$ = 1.62, p = 0.20).

## Discussion

This review of European studies found the prevalence of clinically significant refractive error to be 17.90%, with myopia (≤ −0.50 D), hyperopia (≥ +2.00 D), and astigmatism (≥ 1.00 DC) present in 14.31%, 10.20%, and 10.26% of participants, respectively. Prevalence estimates from cycloplegic studies were lower for myopia and higher for hyperopia than non-cycloplegic studies, although these differences were not statistically significant. Myopia prevalence increased with age, while hyperopia decreased, and astigmatism remained relatively stable. No significant differences in myopia or hyperopia prevalence were observed between males and females. Importantly, variation in the definition of hyperopia across studies substantially influenced reported prevalence rates.

### Clinically significant refractive error prevalence

Nearly one in five children in Europe (17.90%) had clinically significant refractive error. Most previous reviews reported separate figures for myopia and hyperopia, rather than overall refractive error prevalence. This review followed that approach, which likely underestimates true prevalence by excluding astigmatism. Including studies that reported astigmatism raised the estimate to 20.81%. However, due to limited data on how astigmatism co-exists and overlaps with myopia and hyperopia, it couldn't be included in the primary analysis without risking double-counting. Only five countries (Ireland, Northern Ireland, Denmark, Sweden, and Bulgaria) reported data on all three conditions. While all included countries had

# (a)

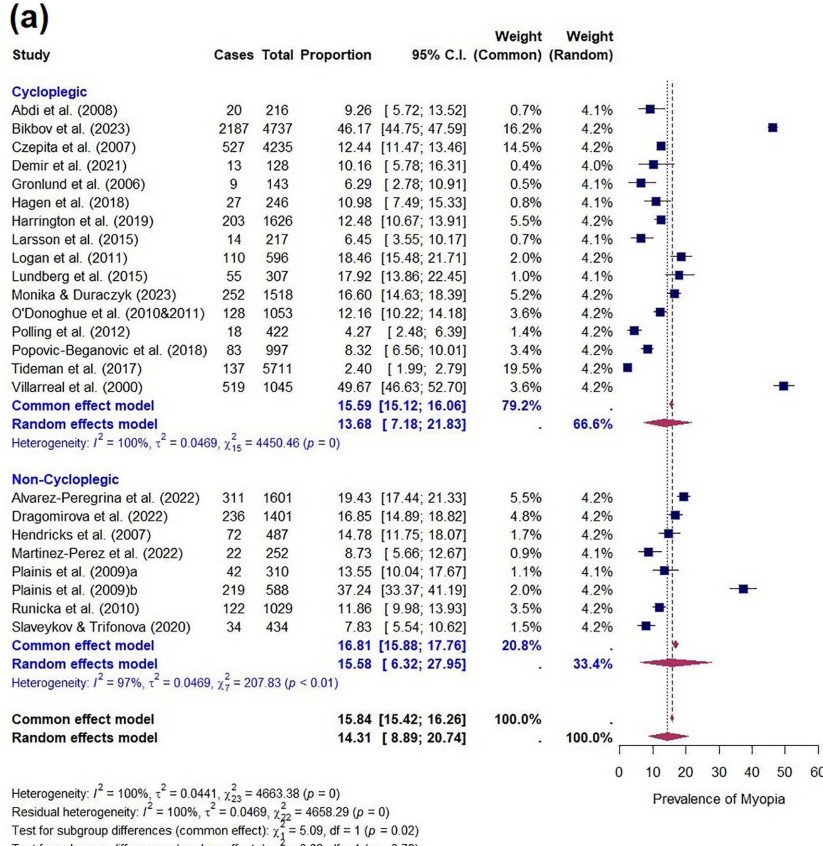

# (b)

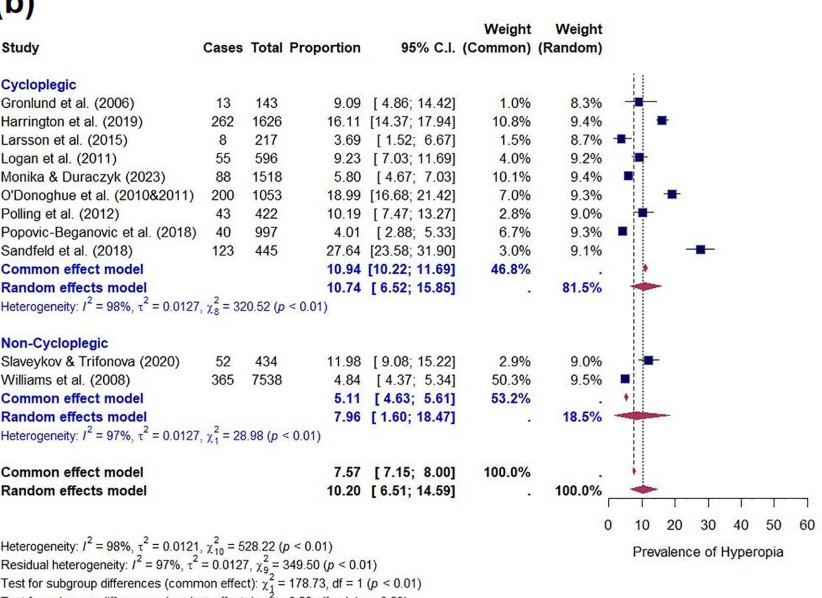

**Fig 5. Forest plot of the prevalence of clinically significant (a) myopia and (b) hyperopia in cycloplegic and non-cycloplegic studies.** Myopia prevalence in cycloplegic studies was 13.68% (95% CI: 7.18–21.83) and 15.58% (95% CI: 6.32–27.95) in non-cycloplegic studies. There was no significant difference in myopia prevalence rates between cycloplegic and non-cycloplegic studies ($\chi^2=0.08$, p=0.78). Hyperopia prevalence in cycloplegic

studies was 10.74% (95% CI: 6.52–15.85) and 7.96% (95% CI: 1.60–18.47) in non-cycloplegic studies. There was no significant difference in hyperopia prevalence rates between cycloplegic and non-cycloplegic studies ($\chi^2 = 0.29$, p = 0.59).

myopia data, fewer had data on hyperopia (n = 8) and astigmatism (n = 6). This highlights a clear gap: Europe urgently needs more consistent, comprehensive data on childhood refractive error.

## Myopia prevalence

Liang et al.'s review reported a higher European paediatric myopia prevalence of 22.60% (95% CI: 17.49–27.72) compared to 14.31% in the present study [10]. However, their estimate included two older studies (1990–2000) with exceptionally high prevalence (EPP: 44.16%) and several others excluded from the present review due to issues such as duplicate data, mismatched age ranges, focus on vision screening failures (which may underestimate true prevalence due to conservative VA thresholds), and unclear cycloplegia use (see S5 Table). These methodological differences likely account for the discrepancy.

A 2021 systematic review of 41 studies from China, Hong Kong, Macau, and Taiwan reported a higher myopia prevalence of 38.0% (95% CI: 35.1–41.1) and lower hyperopia prevalence of 5.2% (95% CI: 3.1–8.6) than observed in European children in the present review [44]. These differences likely reflect ethnic and environmental influences. Myopia is strongly associated with intensive education, prolonged near work, limited outdoor time, and urban living [20,45–47]. Given earlier academic demands and higher urban density in many parts of Asia, elevated myopia rates are expected [48–51].

Globally, Hashemi et al. reported a pooled childhood myopia prevalence of 11.7% based on cycloplegic data from 49 studies (n = 606,155), but their European sample included only studies published between 2000 and 2009 [6]. As prevalence tends to rise in more recent cohorts, this likely contributes to the lower estimate compared with the present review, which includes studies up to June 2024.

## Hyperopia prevalence

Hashemi et al. estimated a global pooled prevalence of childhood hyperopia (≥ + 2.00 D) at 4.6% (95% CI: 3.9–5.2), based on 45 studies involving 200,995 children [6]. Prevalence varied regionally, from 2.2% in Southeast Asia to 14.3% in the Americas. The European estimate was 9.0% (95% CI: 4.3–13.7), drawn from just seven studies in Turkey, Poland, the Netherlands, and Sweden. In contrast, the current review includes additional data from countries such as Denmark (27.64%) [35] and Northern Ireland (18.99%) [15], contributing to a higher pooled European estimate of 10.20%. While Castagno et al. reported hyperopia prevalence ranging from 2.1% to 19.3% across 40 studies in children aged 5–15, they did not provide Europe-specific figures [5]. By incorporating a broader and more recent range of studies, this review offers a more comprehensive estimate of childhood hyperopia prevalence in Europe.

## Astigmatism prevalence

The present review found a prevalence of 10.26% (95% CI: 5.83–15.74) among European children, with the highest rates in Northern Ireland (24% in 6–7-year-olds) and Ireland (19%) [16,36]. In contrast, Hashemi et al. reported a global childhood astigmatism prevalence of 14.9% (95% CI: 12.7–17.1) [6], based on 48 studies and 152,570 participants, using a lower (>0.50 DC) threshold.

Astigmatism may fluctuate over time, with children crossing clinical thresholds during follow-up [20]. Prevalence has also been associated with ethnicity, refractive error type, and educational level [52]. However, only two studies in this review provided data distinguishing myopic from hyperopic astigmatism, limiting further analysis.

## Cycloplegic vs non-cycloplegic comparisons

This meta-analysis found slightly lower myopia prevalence (13.68% vs. 15.58%) and higher hyperopia prevalence (10.74% vs. 7.96%) in cycloplegic compared to non-cycloplegic studies, though differences were not statistically significant. However, no included study assessed both methods within the same population, limiting direct comparison and introducing potential bias from differences in study design, demographics, and methodology. Only two non-cycloplegic studies reported hyperopia prevalence, further restricting comparison.

Nonetheless, external studies consistently demonstrate the importance of cycloplegia. Sankaridurg et al. showed that non-cycloplegic refraction in over 6,000 children in Shanghai significantly overestimated myopia and underestimated hyperopia [53]. Similar findings have been reported elsewhere [54–56], reinforcing the need for cycloplegic assessment in paediatric populations.

In many European countries, however, optometrists are restricted from using diagnostic drops [9], posing a barrier to standardised, cycloplegia-based studies. A multidisciplinary approach may help overcome this challenge and improve the accuracy of epidemiological data.

## Age and sex

The findings of this review align with previous research showing a clear age-related shift in refractive error: hyperopia declines while myopia increases throughout childhood [5,17,57]. In the longitudinal NICER study, myopia prevalence rose from 1.9% to 14.6% in children aged 6–7–12–13, and from 16.4% to 18.6% between ages 12–13 and 18–20. Over the same period, hyperopia in the younger cohort declined from 21.7% to 14.2% [17]. Similarly, Castagno et al. reported a steady drop in hyperopia (≥+2.00D) [5], from 8.4% at age six to around 1% by age 15—consistent with the natural emmetropisation process.

Astigmatism follows a different developmental course. It is high at birth, typically decreases as the cornea flattens, and stabilises by around age four [58,59]. In this review, astigmatism prevalence remained relatively stable (~10%) from ages 4–13, suggesting minimal change beyond early childhood.

Only five studies assessed refractive error in children aged 14 and above, and most focused on myopia and astigmatism. More research is needed in this age group to understand adolescent refractive development and inform future eye care planning and public health policy [60,61].

Sex-based trends remain unclear. While some studies report higher myopia prevalence or progression in females [10,62], others find no difference [63,64]. This review found no significant sex-based differences, but only 12 studies reported sex-disaggregated data. Standardised reporting by sex is needed to enable more robust analyses and guide equitable care strategies.

## Improving generalisability and clarity in prevalence reporting

This review identified no eligible prevalence studies from 27 European countries. Using the UN Geoscheme [65], Northern Europe had the highest representation (13 studies across six countries), while Western Europe was least represented, with only the Netherlands included. This lack of data prevented meaningful regional subgroup analysis, as single-country data cannot reflect broader regional trends. More studies from under-represented areas are needed to improve geographic balance and generalisability in future meta-analyses.

Clinically significant hyperopia was defined as ≥+2.00D. Nine studies using lower thresholds were excluded to avoid overestimating prevalence by including children with age-appropriate hyperopic reserve. While measuring hyperopic reserve is valuable—especially due to its protective role against myopia [66,67], lower thresholds hinder comparability across studies. Standardised definitions are essential for consistent epidemiological reporting [14].

Most studies used spherical equivalent refraction (SER), which may misclassify children with high astigmatism. This can skew prevalence estimates, underestimating hyperopia and overestimating myopia. Alternative methods, such as least myopic meridian or vector analysis, may offer more accurate classification [2,68].

**Strengths, limitations and future directions**

This systematic review and meta-analysis, conducted in line with PRISMA guidelines, provides the first comprehensive estimate of childhood refractive error prevalence across Europe. A key strength is identifying critical data gaps—specifically, the absence of eligible studies in 27 European countries. Rather than a limitation, this highlights urgent research needs in under-represented regions. Inconsistencies in study protocols and refractive error definitions also posed challenges, reinforcing the importance of standardisation in future research.

The broad timeframe and heterogeneous cohorts may have influenced the pooled estimates. This is an inherent feature of prevalence meta-analyses, and the use of a random-effects model with sensitivity analyses was intended to account for this [69–71].

This review presents the first comparative analysis of cycloplegic versus non-cycloplegic prevalence estimates. While direct within-study comparisons were unavailable, the findings offer important insights into the influence of cycloplegia on prevalence data. Subgroup analyses by age and sex further strengthen the review, although the age-based findings are drawn mainly from cross-sectional data. With only one longitudinal dataset (NICER) included, future meta-analyses incorporating longitudinal data could offer a clearer picture of refractive error progression and its determinants [72]. Future studies should standardise methods and expand coverage to under-represented regions, enabling more accurate and comparable prevalence estimates.

## Conclusions

This meta-analysis provides the first comprehensive estimates of clinically significant refractive error among European children, with an overall prevalence of 17.9%, including 14.3% for myopia, 10.2% for hyperopia, and 10.3% for astigmatism. It also demonstrates the impact of cycloplegia on prevalence estimates and highlights critical data gaps across much of Europe.

To ensure accurate, comparable, and actionable data, there is an urgent need for globally coordinated efforts to standardise study protocols, definitions, and reporting. Myopia, hyperopia, and astigmatism each carry distinct clinical and public health implications. Consistent, granular data will enable more targeted screening, optimised resource allocation, and more effective eye care strategies in Europe and worldwide.

## Supporting information

**S1 Appendix. Protocol for the meta-analysis.**
(PDF)

**S1 Fig. Leave-one-out sensitivity plot of all studies reporting myopia or hyperopia prevalence in Europe.** Leave-one-out analyses were performed to determine how much individual studies affect the pooled estimates of the other studies.
(TIF)

**S2 Fig. Forest plots displaying the prevalence of clinically significant (a) myopia (≤−0.50D), (b) hyperopia (≥+2.00D) and (c) astigmatism (≥1.00 DC) in 4–17-year-olds in Europe across 27 studies (37,282 participants, 14 countries, 18 cycloplegic, 9 non-cycloplegic).** Myopia prevalence was 14.31% (95% CI: 8.89–20.74). Hyperopia prevalence was 10.20% (95% CI: 6.51–14.59). Astigmatism prevalence was 10.26% (95% CI: 5.83–15.74).
(TIF)

**S3 Fig. Heat maps of clinically significant paediatric refractive error prevalence by country.** Figure 3(a) depicts myopia (≤−0.50D) prevalence, (b) hyperopia (≤+2.00D) prevalence and (c) astigmatism (≥1.00 DC) prevalence. Mapping data from http://www.naturalearthdata.com/.
(TIF)

**S1 Table. PRISMA-P (Preferred Reporting Items for Systematic review and Meta-Analysis Protocols) 2015 checklist: recommended items to address in a systematic review protocol.**
(DOCX)

**S2 Table. MOOSE (Meta-analyses Of Observational Studies in Epidemiology) checklist.**
(PDF)

**S3 Table. Critical appraisal of included papers using the Joanna Briggs Institute tool.**
(DOCX)

**S4 Table. Grading of Recommendations, Assessment, Development and Evaluation (GRADE) summary of findings.**
(DOCX)

**S5 Table. Excluded papers and reason for exclusion.**
(DOCX)

## Author contributions

**Conceptualization:** Michael Moore.

**Data curation:** Megan Doyle, Aoife O'Donnell.

**Formal analysis:** Megan Doyle, Michael Moore.

**Investigation:** Megan Doyle, Aoife O'Donnell.

**Methodology:** Megan Doyle, Aoife O'Donnell, Siofra Harrington, Veronica O'Dwyer, Michael Moore.

**Supervision:** Siofra Harrington, Veronica O'Dwyer, Michael Moore.

**Validation:** Megan Doyle, Aoife O'Donnell, Siofra Harrington, Michael Moore.

**Visualization:** Megan Doyle, Siofra Harrington, Michael Moore.

**Writing – original draft:** Megan Doyle.

**Writing – review & editing:** Aoife O'Donnell, Siofra Harrington, Veronica O'Dwyer, Michael Moore.

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
