## [Decision Letter · Decision Letter 0]

20 Aug 2025

Thank you for submitting your manuscript to PLOS ONE. After careful consideration, we feel that it has merit but does not fully meet PLOS ONE’s publication criteria as it currently stands. Therefore, we invite you to submit a revised version of the manuscript that addresses the points raised during the review process.

We look forward to receiving your revised manuscript.

Kind regards,

Neelam Pawar

Academic Editor

PLOS ONE

**Journal Requirements:**

1. When submitting your revision, we need you to address these additional requirements. Please ensure that your manuscript meets PLOS ONE's style requirements, including those for file naming. The PLOS ONE style templates can be found at https://journals.plos.org/plosone/s/file?id=wjVg/PLOSOne_formatting_sample_main_body.pdf and https://journals.plos.org/plosone/s/file?id=ba62/PLOSOne_formatting_sample_title_authors_affiliations.pdf 2. Thank you for stating the following in the Acknowledgments Section of your manuscript: This work was supported by the Technological University Dublin Fiosraigh Scholarship and Optometry Ireland. We note that you have provided funding information that is not currently declared in your Funding Statement. However, funding information should not appear in the Acknowledgments section or other areas of your manuscript. We will only publish funding information present in the Funding Statement section of the online submission form. Please remove any funding-related text from the manuscript and let us know how you would like to update your Funding Statement. Currently, your Funding Statement reads as follows: The author(s) received no specific funding for this work.  Please include your amended statements within your cover letter; we will change the online submission form on your behalf. 3. Thank you for stating the following in the Competing Interests section: I have read the journal's policy and the authors of this manuscript have the following competing interests: MM is a consultant for Thea Pharmaceuticals We note that one or more of the authors are employed by a commercial company: Thea Pharmaceuticals  a. Please provide an amended Funding Statement declaring this commercial affiliation, as well as a statement regarding the Role of Funders in your study. If the funding organization did not play a role in the study design, data collection and analysis, decision to publish, or preparation of the manuscript and only provided financial support in the form of authors' salaries and/or research materials, please review your statements relating to the author contributions, and ensure you have specifically and accurately indicated the role(s) that these authors had in your study. You can update author roles in the Author Contributions section of the online submission form. Please also include the following statement within your amended Funding Statement. “The funder provided support in the form of salaries for authors [insert relevant initials], but did not have any additional role in the study design, data collection and analysis, decision to publish, or preparation of the manuscript. The specific roles of these authors are articulated in the ‘author contributions’ section.”If your commercial affiliation did play a role in your study, please state and explain this role within your updated Funding Statement.  b. Please also provide an updated Competing Interests Statement declaring this commercial affiliation along with any other relevant declarations relating to employment, consultancy, patents, products in development, or marketed products, etc.   Within your Competing Interests Statement, please confirm that this commercial affiliation does not alter your adherence to all PLOS ONE policies on sharing data and materials by including the following statement: "This does not alter our adherence to  PLOS ONE policies on sharing data and materials.” (as detailed online in our guide for authors http://journals.plos.org/plosone/s/competing-interests) . If this adherence statement is not accurate and  there are restrictions on sharing of data and/or materials, please state these. Please note that we cannot proceed with consideration of your article until this information has been declared. Please include both an updated Funding Statement and Competing Interests Statement in your cover letter. We will change the online submission form on your behalf. 4. We note that Figures 4 and S3 in your submission contain map images which may be copyrighted. All PLOS content is published under the Creative Commons Attribution License (CC BY 4.0), which means that the manuscript, images, and Supporting Information files will be freely available online, and any third party is permitted to access, download, copy, distribute, and use these materials in any way, even commercially, with proper attribution. For these reasons, we cannot publish previously copyrighted maps or satellite images created using proprietary data, such as Google software (Google Maps, Street View, and Earth). For more information, see our copyright guidelines: http://journals.plos.org/plosone/s/licenses-and-copyright. We require you to either present written permission from the copyright holder to publish these figures specifically under the CC BY 4.0 license, or remove the figures from your submission: a. You may seek permission from the original copyright holder of Figures 4 and S3 to publish the content specifically under the CC BY 4.0 license.   We recommend that you contact the original copyright holder with the Content Permission Form (http://journals.plos.org/plosone/s/file?id=7c09/content-permission-form.pdf) and the following text:“I request permission for the open-access journal PLOS ONE to publish XXX under the Creative Commons Attribution License (CCAL) CC BY 4.0 (http://creativecommons.org/licenses/by/4.0/). Please be aware that this license allows unrestricted use and distribution, even commercially, by third parties. Please reply and provide explicit written permission to publish XXX under a CC BY license and complete the attached form.” Please upload the completed Content Permission Form or other proof of granted permissions as an "Other" file with your submission. In the figure caption of the copyrighted figure, please include the following text: “Reprinted from [ref] under a CC BY license, with permission from [name of publisher], original copyright [original copyright year].” b. If you are unable to obtain permission from the original copyright holder to publish these figures under the CC BY 4.0 license or if the copyright holder’s requirements are incompatible with the CC BY 4.0 license, please either i) remove the figure or ii) supply a replacement figure that complies with the CC BY 4.0 license. Please check copyright information on all replacement figures and update the figure caption with source information. If applicable, please specify in the figure caption text when a figure is similar but not identical to the original image and is therefore for illustrative purposes only.The following resources for replacing copyrighted map figures may be helpful: USGS National Map Viewer (public domain): http://viewer.nationalmap.gov/viewer/The Gateway to Astronaut Photography of Earth (public domain): http://eol.jsc.nasa.gov/sseop/clickmap/Maps at the CIA (public domain): https://www.cia.gov/library/publications/the-world-factbook/index.html and https://www.cia.gov/library/publications/cia-maps-publications/index.htmlNASA Earth Observatory (public domain): http://earthobservatory.nasa.gov/Landsat:
http://landsat.visibleearth.nasa.gov/USGS EROS (Earth Resources Observatory and Science (EROS) Center) (public domain): http://eros.usgs.gov/#Natural Earth (public domain): http://www.naturalearthdata.com/ 5. If the reviewer comments include a recommendation to cite specific previously published works, please review and evaluate these publications to determine whether they are relevant and should be cited. There is no requirement to cite these works unless the editor has indicated otherwise. 

Reviewers' comments:

**Comments to the Author**

1. Is the manuscript technically sound, and do the data support the conclusions?

Reviewer #1: Partly

Reviewer #2: Yes

2. Has the statistical analysis been performed appropriately and rigorously?

Reviewer #1: Yes

Reviewer #2: Yes

3. Have the authors made all data underlying the findings in their manuscript fully available?

Reviewer #1: Yes

Reviewer #2: Yes

4. Is the manuscript presented in an intelligible fashion and written in standard English?

Reviewer #1: Yes

Reviewer #2: Yes

**Reviewer #1:**  Dear Authors,

I have few concerns regarding the data accuracy. You included in your analysis a recent study from Russia (Bibkov&All, 2023) on a very consistent group of subjects. I red the abstract of this study and actually there is no mention in this study about Hyperopia incidence (in your table appears as being 11.1%) because it is a study focused exclusively on Myopia.

From the clinical point of vu, I think your analysis is expanded on a very large period and the cohorts included in the studies are very heterogeneous and the large groups influence the final statistics.

**Reviewer #2:**  General comments:

The article is well written, with a clearly defined PICO framework and well-articulated aims. The methodology meets the criteria for a meta-analysis, with appropriate use of plots and figures to present the findings. The conclusions are clearly stated and consistent with the results.

Minor comments :

1- Introduction, line 50

Consider specifying “uncorrected refractive error” instead of “refractive error” to be more precise, as the burden on visual impairment is primarily due to uncorrected cases.

2- Introduction, line 57

The first sentence could be rephrased for clarity. For example: “While severe refractive error carries greater individual risk, the broader public health burden stems from the vast number of people with low to moderate refractive error, particularly myopia.”

**Do you want your identity to be public for this peer review?** For information about this choice, including consent withdrawal, please see our Privacy Policy

Reviewer #1: No

Reviewer #2: No

---

## [Author Response · Author response to Decision Letter 1]

24 Sep 2025

Dr Emily Chenette, PhD

Editor-In-Chief

PLoS One

11th September 2025

Dear Doctor Chenette,

Thank you for the opportunity to submit our revised manuscript entitled "Prevalence of Clinically Significant Refractive Error in Children in Europe: Systematic Review and Meta-Analysis" for consideration for publication in PLoS One.

The authors thought the feedback was positive, with helpful pointers for the paper. The authors appreciate the time and effort the reviewers have dedicated to providing valuable feedback on the manuscript. The authors have been able to incorporate changes to reflect the suggestions provided by the reviewers.

In summary, having considered the comments in some detail, the following amendments have been made:

1. Two short sections have been added to clarify points highlighted by Reviewer 1.

2. Lines 50-51, and lines 57 have been rephrased as per Reviewer 2 comments.

The authors have highlighted the changes within the manuscript using tracked changes. Here is a point-by-point response, including edits (italicised) to the reviewers' comments and concerns (in boldface).

Reviewer: 1

Dear Authors,

I have few concerns regarding the data accuracy. You included in your analysis a recent study from Russia (Bikbov et al., 2023) on a very consistent group of subjects. I read the abstract of this study and actually there is no mention in this study about Hyperopia incidence (in your table appears as being 11.1%) because it is a study focused exclusively on Myopia. From the clinical point of view, I think your analysis is expanded on a very large period and the cohorts included in the studies are very heterogeneous and the large groups influence the final statistics.

The authors thank the reviewer for their thoughtful comments and the opportunity to clarify these points.

With regard to the concern about the Russian study by Bikbov et al.,1 while hyperopia prevalence is not mentioned in the abstract, it is reported in the main body of the paper. Specifically, in the results section (page 594), the authors state: “… and the prevalence of hyperopia (>0.50 D) was 524/4737 (11.1%; 95% CI 10.2–12.0%).” The value reported in Table 2 of the current manuscript accurately reflects this finding.

On the issue of study periods, there is currently limited refractive error prevalence data available for children in Europe. Restricting the timeframe would have reduced the evidence included in the current analysis. Previous meta-analyses of myopia prevalence have adopted similar inclusion periods; for example, Liang et al.2 analysed studies from 1990-2023, while Hashemi et al.3 included data from 1990-2016. The approach used in the current meta-analysis therefore aligns with established practice and extends the evidence base by incorporating refractive error types beyond myopia.

This was clarified in the manuscript as follows:

2.1 Literature Search Strategy, Lines 87-92: “To maximise the available evidence, studies published from 1 January 2000 to 30 September 2024 were included. This broad timeframe was consistent with previous prevalence meta-analyses and ensured the inclusion of recent, methodologically robust studies while allowing adequate data coverage for European children, where studies remain limited.5,6,10”

The authors agree with the reviewer that heterogeneity is an inherent feature of prevalence meta-analyses, reflecting variations in study populations, geography, methodology, sample size, and timeframes.4 To address this, a random-effects model was applied, which is designed to account for both within- and between-study variability.5 Additionally, Freeman-Tukey double arcsine transformation and leave-one-out sensitivity analyses were employed to reduce the influence of outlying prevalence estimates and to ensure that larger studies did not disproportionately affect the pooled results.6 Taken together, these methodological choices follow established best practices and provide robust estimates despite the unavoidable heterogeneity in the available data.

This was clarified in the manuscript as follows:

4.8 Strengths, Limitations and Future Directions, Lines 445-447: “The broad timeframe and heterogeneous cohorts may have influenced the pooled estimates. This is an inherent feature of prevalence meta-analyses, and the use of a random-effects model with sensitivity analyses was intended to account for this.69-71”

References:

1. Bikbov MM, Kazakbaeva GM, Fakhretdinova AA, Tuliakova AM, Iakupova EM, Panda-Jonas S, et al. Prevalence and associated factors of myopia in children and adolescents in Russia: the Ural Children Eye Study. BJO. 2024;108:593-598.

2. Liang J, Pu Y, Chen J, Liu M, Ouyang B, Jin Z, et al. Global prevalence, trend and projection of myopia in children and adolescents from 1990 to 2050: A comprehensive systematic review and meta-analysis. BJO. 2024;109(3):362-71.

3. Hashemi H, Fotouhi A, Yekta A, Pakzad R, Ostadimoghaddam H, Khabazkhoob M. Global and regional estimates of prevalence of refractive errors: Systematic review and meta-analysis. J Curr Ophthalmol. 2018;30:3–22.

4. Migliavaca CB, Stein C, Colpani V, Barker T, Ziegelmann P, Munn Z, et al. Meta-analysis of prevalence: I2 statistic and how to deal with heterogeneity. Res Synth Methods. 2022;13(3):363-367.

5. Borenstein M, Hedges LV, Higgins JPT, Rothstein HR. Random-Effects Model. In: Introduction to Meta-Analysis. Wiley;2009:69-75.

6. Barendregt JJ, Doi SA, Lee YY, Norman RE, Vos T. Meta-analysis of prevalence. J Epidemiol Community Health. 2013;67:974-978.

Reviewer: 2

General comments:

The article is well written, with a clearly defined PICO framework and well-articulated aims. The methodology meets the criteria for a meta-analysis, with appropriate use of plots and figures to present the findings. The conclusions are clearly stated and consistent with the results.

The authors thank the reviewer for their positive feedback.

Minor comments:

1- Introduction, line 50

Consider specifying “uncorrected refractive error” instead of “refractive error” to be more precise, as the burden on visual impairment is primarily due to uncorrected cases.

The authors thank the reviewer for this suggestion. The introduction has been updated to reflect this as follows:

Lines 50-51: “Uncorrected refractive error is a leading cause of visual impairment worldwide and a growing public health concern, particularly in children.”

2- Introduction, line 57

The first sentence could be rephrased for clarity. For example: “While severe refractive error carries greater individual risk, the broader public health burden stems from the vast number of people with low to moderate refractive error, particularly myopia.”

The authors thank the reviewer for their comment. The introduction has been updated to incorporate this as follows:

Lines 57-59: “While severe refractive error carries greater individual risk, the broader public health burden stems from the vast number of people with low to moderate levels of refractive error, particularly myopia.”

Editor Comments:

The manuscript has been updated to follow the style requirements. The competing interests and funding statements have been updated as requested. Figure 4 and S3 have been recreated using public domain data which is cited in the figure legend.

The authors thank the editorial board and the reviewers for their constructive comments. We appreciate your input, which has improved the paper's clarity and readability.

Yours sincerely,

Michael Moore (corresponding author), Centre for Eye Research Ireland, School of Physics, Clinical & Optometric Sciences, Technological University Dublin, City Campus, Grangegorman, Dublin, Ireland. Email: michael.moore@tudublin.ie

---

## [Editor Report · Decision Letter 1]

15 Oct 2025

Prevalence of Clinically Significant Refractive Error in Children in Europe: Systematic Review and Meta-Analysis

PONE-D-25-23846R1

Dear Dr.Michael Moore

We’re pleased to inform you that your manuscript has been judged scientifically suitable for publication and will be formally accepted for publication once it meets all outstanding technical requirements.

Kind regards,

Neelam Pawar

Academic Editor

PLOS ONE
---

## [Editor Report · Acceptance letter]

PONE-D-25-23846R1

PLOS ONE

Dear Dr. Moore,

I'm pleased to inform you that your manuscript has been deemed suitable for publication in PLOS ONE. Congratulations! Your manuscript is now being handed over to our production team.

Kind regards,

on behalf of

Dr. Neelam Pawar

Academic Editor

PLOS ONE